# A Multi-Layer Semantic Approach for Digital Forensics Automation for Online Social Networks

**DOI:** 10.3390/s22031115

**Published:** 2022-02-01

**Authors:** Humaira Arshad, Saima Abdullah, Moatsum Alawida, Abdulatif Alabdulatif, Oludare Isaac Abiodun, Omer Riaz

**Affiliations:** 1Department of Computer Sciences, The Islamia University of Bahawalpur, Bahawalpur 63100, Pakistan; saima.abdullah@iub.edu.pk; 2Department of Computer Sciences, Abu Dhabi University, Abu Dhabi 59911, United Arab Emirates; 3Department of Computer Science, College of Computer, Qassim University, Buraydah 51452, Saudi Arabia; ab.alabdulatif@qu.edu.sa; 4Department of Computer Science, University of Abuja, Gwagwalada 900110, Nigeria; oludare.abiodun@uniabuja.edu.ng; 5Department of Information Technology, Faculty of Computing, The Islamia University of Bahawalpur, Bahawalpur 63100, Pakistan; omer.riaz@iub.edu.pk

**Keywords:** forensic applications, social network forensics, forensic automation, automation tools, experimental visualization, semantic data presentation, evidence analysis

## Abstract

Currently, law enforcement and legal consultants are heavily utilizing social media platforms to easily access data associated with the preparators of illegitimate events. However, accessing this publicly available information for legal use is technically challenging and legally intricate due to heterogeneous and unstructured data and privacy laws, thus generating massive workloads of cognitively demanding cases for investigators. Therefore, it is critical to develop solutions and tools that can assist investigators in their work and decision making. Automating digital forensics is not exclusively a technical problem; the technical issues are always coupled with privacy and legal matters. Here, we introduce a multi-layer automation approach that addresses the automation issues from collection to evidence analysis in online social network forensics. Finally, we propose a set of analysis operators based on domain correlations. These operators can be embedded in software tools to help the investigators draw realistic conclusions. These operators are implemented using Twitter ontology and tested through a case study. This study describes a proof-of-concept approach for forensic automation on online social networks.

## 1. Introduction

Forensic data extraction on social networks has emerged as a significant research issue [1]. The forensic data collection is inextricably linked to the service operator, resulting in data authenticity and formatting issues. Social network forensics is an emerging domain of digital forensics [2]. These platforms record an assortment of data related to an individual’s social behaviors, associations, and preferences. Several studies explore this information to analyze business predictions, health tendencies, and psychological insights. With online social networks, a complete set of all user communications is stored at the service provider, with no direct access for investigators. Some traces of these communications are stored on the smart devices used to access these platforms, although they offer a partial or fractured view of these communications. However, in legal proceedings, the suspect and victim are compelled by court order to provide complete access to their online data to investigators. The present abundance of digital devices combined with excessive use by individuals produces immense digital data volumes every day on social media platforms and handheld devices [3]. In this paper, we identify critical data sources and several automated analytical techniques for forensic analysis of social network user data to examine and investigate these data to determine a crime.

This study identifies and examines the critical issues that must be addressed for practical automated solutions. It classifies these issues into two significant categories, namely technical and legal issues.

### 1.1. Technical Issues

Technical issues arise due to social media networks’ enormous size, distributed nature, and diversity.

#### 1.1.1. Automated Collection

Automated collection processes are well established in device-based digital forensics. However, online social networks, cloud, and IoT systems offer varied challenges to the well-established collection procedures mainly because distributed and open-ended design heterogeneity of data and privacy laws [3,4]. These systems mostly need automated collection and preservation methods to handle the massive amount of data quickly and correctly. However, an automated extraction requires clearly defined extraction boundaries. Hence, a precisely outlined investigative procedure for OSNs (online social networks) is needed to address all the crucial aspects of the process.

#### 1.1.2. Automated Analysis

OSN forensic tools currently use only keyword searches, which is not enough for current social media investigations due to social media content’s enormous size and intricate nature. This content includes text, reactions, and multimedia content in images and videos. This fact is also acknowledged by [5,6]. The investigator should filter the data with a customized and advanced querying mechanism to gain an insight into a particular sequence of events, such as provided by event-based models in computer forensics; these models are explained in [3,6,7]. These models explain the analysis process logically; hence, they provide a theoretical background for forensic analysis.

### 1.2. Legal Issues

The result and conclusions of a forensic process are presented in a court of law as evidence; hence, the forensic process must adhere to judicial requirements and constraints. Otherwise, the evidence is rejected in judicial proceedings [8]. This study examines the two key legal issues, besides privacy issues, in the online social network domain that are specifically relevant for achieving automation: data provenance and formal theory.

#### 1.2.1. Data Provenance

The record describing the origin and other historical information about a piece of data is referred to as its provenance. An advanced data provenance system will provide forensic investigators with a clear picture of the lineage of the data. It will aid in the resolution of disputes over sensitive data by providing digital evidence. Secure provenance, which records data objects’ ownership and process history, is critical to cloud computing forensic investigation success.

Only methods and techniques that can establish data provenance and describe the automatically generated results using a logically explainable theory are accepted [9,10]. Several data mining techniques lose provenance information while pre-processing and normalizing the data [11,12] As a result, most methods for analyzing social media using data mining techniques are not approved for legal or forensic use because they tend to lose provenance during data pre-processing and processing [13,14,15,16,17,18].

#### 1.2.2. Formal Theory

In scientific disciplines, including forensic sciences, formal theories explain the process and reasoning behind any [19,20,21]. Formal theory is a mathematical model or set of rules and assumptions used to understand various behaviors. Gladyshev proposed an approach based on a finite state machine to explain the logical order of an event on a computer machine [19]. Likewise, Somayeh Soltani used a temporal logical model to order the events involved in a crime [20]. J. Wang proposed a timed automata technique to reconstruct the potential events in cloud forensics [21].

In a scenario where two suspects have the option to confess or refuse to speak about a potential crime, the model employs a set of rules to determine the outcomes of various behaviors. It is written, read, analyzed, and debated in social work and academia. It aids in understanding and expanding our understanding of the world around us by providing an enormously powerful set of tools for exploration and explanation. Unfortunately, such standard approaches are very few in digital forensics, mainly due to the rapid evolution of technology and heterogeneity of devices and operating systems. Hence, the forensic conclusions based on data analysis in the absence of proven theories are not treated as scientific evidence. A formalized knowledge model of the relevant domain can help demonstrate and interpret the automated results.

## 2. Related Works

The term “Social Media Forensics” refers to a subset of Network Forensics, which refers to the process of retrieving evidence from online social networking activities. Various social sites, such as Facebook, Myspace, Twitter, and LinkedIn, are common among internet users. The following fourteen sections are reflected in this paper: proposed approach, implementation, data collection, ontologies and RDF, data stores, mapping and instantiation, knowledge enhancement, knowledge extraction, automated analysis operators, visualizations, case study, test data, experiments, results, and limitations.

Forensic science is undergoing a transformation and expansion to include modern evidence such as digital evidence. However, the process is causing several operational and conceptual challenges. The papers by Biedermann and Kotsoglou [22] and Stoyanova et al. [4] reviewed and discussed a series of convoluted conceptual hurdles experienced in linkage with the use of digital evidence as part of the evidence and proof processes at trial. The significant artefacts needed in forensic examination and collection are diverse and dissimilar from the device-based analysis in online networks.

Social network data encompass numerous users, several machines, and a few time zones. Notably, social networks are designed using diverse structure and service models and various data formats. Hence, the existing approaches are not appropriate for data collection and evidence examination from OSNs, as explained by other researchers in the domain [23,24,25,26,27]. Existing approaches fundamentally provide a few standard guidelines to the investigators for data collection and analysis. Some methods have offered investigative techniques customized for OSNs in recent years [28,29,30]. Jadhao presented an ontology-based semantic method to identify criminal activity using specific words [23]. N. Zainudin presented a digital forensic process model for investigating online social networks. The work emphasized the necessity of automated techniques to deal with OSN data but does not offer any concrete method [29]. The same approach with slightly more details is presented in their subsequent work [30]. Although these approaches provide minimum assistance, they do not focus on all the crucial facets of social network forensic investigation, such as specifying incidents precisely, defining crime scene boundaries, and establishing provenance.

Many forensic tools cannot process advanced analysis features due to the underlying unstructured data, which lack any semantic information required for automated reasoning. This situation could be helped with the structured and formal representation of data, which will support the smooth development of automated processes and be easy to understand for investigators when presented through graphs, visualization tools, and query tools. Few approaches exist for representing and exchanging cyber-investigation data when combining data sources from diverse organizations or dealing with large amounts of data from multiple tools. Casey et al. [31] used an open community-developed specification language called Cyber-Investigation Analysis Standard Expression (CASE) to address this need for information exchange and tool interoperability. The outcome demonstrated a proof-of-concept Application Program Interface (API) to facilitate CASE tool implementation. Thus, community members can develop and implement CASE and the Unified Cyber Ontology (UCO) [31]. However, these approaches are focused on unifying and structuring the evidence data for exchange purposes. They are not addressing the necessity of organizing the data in a structured and quarriable format for analysis objectives. Mainly, the reason is criticism of automated forensic analysis tools and their lack of acceptance in the judicial system.

Content analysis on OSNs uses data mining approaches to identify a suspect or predict a [13,14,15,16,17]. Several approaches use statistical and probability-based methods [11,12]. However, due to these methods and pre-processing mechanisms, they lose the data provenance during computations. Provenance refers to the source and record of an object. Establishing and managing the provenance of any item intended to use as evidence is crucial in forensics as it equates to the chain of custody. Thus, the metadata surrounding a digital object are recorded to demonstrate the authenticity of an item. These metadata include information such as the time of creation, the user’s identity, and modifications.

Otherwise, the data processed through unexplainable methods that could not demonstrate the data origin, such as data mining methods, are not acceptable in legal proceedings. Although data mining techniques suitably show criminal activities in automated systems, they do not offer roofs that can be traced to their origin, which is essentially needed in judicial proceedings [15]. Hence, every automated technique or procedure must satisfy the judicial criteria for gaining acceptance and admissibility of the proofs.

Hence, the current searching and sorting features needed for data analysis are limited to keyword search due to the storage formats and underlying data organization [5,32]. Digital forensic analysis tools in practice include file viewers, file analysis tools, registry analysis tools, internet analysis tools, and email analysis tools. Mobile device analysis tools include specialized tools such as Encase, CacheBack, and Internet Evidence Finder (IEF), commonly used in investigative practice [33,34]. Some tools deployed for data analysis include AXIOM, autopsy^®^, and Magnet ACQUIRE. Axiom is a full-service investigation platform that can recover, analyze, and report on data from mobile, computer, and cloud sources in a single case file.

In comparison, autopsy^®^ is a digital forensics platform that also serves as a graphical interface for The Sleuth Kit^®^ and other digital forensics tools. It is used to investigate what happened on a computer by law enforcement, military, and corporate examiners. It can even be used to recover photos from the camera’s memory card. Meanwhile, Magnet ACQUIRE is a free tool for digital forensic examiners that allows them to acquire forensic images of any iOS or Android device quickly and easily from the hard drive or removable media. Furthermore, due to the wide range of different types of computer-based evidence, a variety of computer forensics tools are available, including disk and data capture tools.

In other forensic disciplines, formal theories are used to explain the conclusions. However, few such approaches have been presented in digital forensics, and they are not suitable for OSN. The most prominent are presented by Gladyshev, Carrier, Cohen, and Chabot [7,24,35,36,37]. Carrier and Spafford presented the idea of atomic and complex events, and they proposed to integrate the digital crime scene with a physical crime scene [24]. Cohen emphasized the necessity of formal theories in digital forensics and suggested using mathematical terminologies and methods to avoid inconsistencies [36]. Gladyshev used a finite state machine approach to reconstruct the potential order of events in a crime conducted through a personal computer [13]. Chabot presented a theoretical approach based on temporal differences and overlaps to identify the chronological order of events around a crime [36].

Therefore, in social media forensics software, even the tools designed explicitly to facilitate OSN forensics such as X1 Social discovery and many other tools such as Aleph Archives, NextPoint, and Hanzo Archives provide data sorting and keyword searching features [38]. Their analysis capabilities are also restricted due to the linear storage formats Such as Aleph Archives and Hanzo Archives store data in the WARC Web archive format; X1 saves data in the MHT Web archive format and exports to Concordance, CSV, and HTML. NextPoint stores data as PDF, HTML, and PNG files; it also exports data to Concordance and XML [39]. Most of these formats are textual and linear. However, we noted that these tools are suitable for quicker and more extensive data extraction and proper preservation to warrant data integrity. Hence, they are suitable only for forensic acquisition and early case assessment.

Automation seems a reasonable solution to handle the heterogeneous, distributed, and massive social network forensics data sources. The automation of forensic techniques and processes is not entirely a technical issue. The evidence obtained due to automated methods is meant to be used in a court of law. Digital forensics methods are not valuable from a legal perspective if they fail to qualify judicial admittance standards. Fully automated forensic systems are sharply criticized both by legal and academic communities. However, presently, automation is crucial to managing massive data loads, as acknowledged by several researchers and critics [8,37,40]. The automation process can increase the investigative process’s efficiency by increasing the speed and reducing the time required to collect and analyze data. However, the researchers and legal practitioners insist that automation should be carefully applied in few specific phases to some extent. Another study published design requirements for automated forensic tools. The work also provides a hierarchy for evaluating an automated forensics system [41].

## 3. Proposed Approach

This study examines the potential areas of a forensics investigation where automation can be applied without contradicting online social networks’ legal and privacy requirements. We propose an automation model to address the automation issues at several social network forensics phases. We identify the key process areas for automation and classify them into distinct layers, suggesting appropriate automated solutions for each layer, as shown in Figure 1. The aim is to provide automated analysis methods to perform complex and logical queries on the data and metadata gathered from OSNs. However, a structured or at least linked data representation is necessary to formulate such sophisticated analysis methods. Hence, the other key area is data organization. As the analysis operator is intended to identify and reveal an otherwise unrelated aspect of data, they need advanced visualizations of data; simple representation of tabular data would fail to show any trend or pattern existing in the data. This issue is addressed in the interface layer. The legal requirements for justifying and explaining the analysis operators demand an explainable theory explained in the knowledge layer. We propose separate models for managing evidence collection, analysis, and interpretation. We also propose a five-layer model to support forensic automation on OSN. The layered model is outlined in the following diagram (Figure 1). The details of each layer are explained in the following paragraphs.

The knowledge layer is the lowest layer that correctly interprets social network contents in the forensic context. This theory is meant to justify the legal and technical requirements to explain the facts drawn as evidence. It explains the logical sequence and reasoning used to extract a fragment of OSN content as evidence. The theory’s formalism would help interpret and verify the proofs obtained by automated methods in a court of law.

The process layer highlights the critical process areas for automation. These process areas include incident identification and evidence acquisition. This layer is also responsible for stating the parameters such as the scope and type of data extraction, specifying significant actors involved in an incident. A precise specification of incident parameters is required to automate the complete investigation procedure. An investigation process model for online social networks forensics was proposed to address this layer’s requirements in a previous study [42].

The data layer provides simplified access to the data related to the incident and extracted from OSNs. The data layer implements the knowledge model using semantic web methods and ontologies. Therefore, this layer is responsible for normalizing the obtained data and saving them in persistent storage. The persistent storage is designed and implemented using the semantic web schema and RDF stores.

The analysis of such varied and unstructured datasets requires advanced and customized tools. Hence, the analysis layer is meant to represent analysis operators to perform automated analysis in the context of social networks. In this article, we propose a few computerized analysis methods to quickly sort and analyze the data and present the results to the human examiners for evaluation. The decision-making process is delegated to human examiners due to many scenarios and various crimes investigated through social network evidence.

The interface layer permits the investigators to interact with the data and analysis layer. This layer aims to present the processed data in a manner that is instantly understandable by a non-technical person. The results generated by analysis operators are presented using suitable visualizations such as heat maps and activity diagrams, relative frequency, and cumulative frequency histograms. Figure 2 provides a complete architecture for the implementation of the multi-layered system.

Figure 2 also explains the association among the models.

## 4. Implementation

The layered approach proposed in the previous section is implemented through separate models at each layer. An overview of the techniques and tools used for implementing the proposed model is given in Figure 3.

The knowledge model formulated in this work is an event-based model that is exclusively prepared for electronic forensics and analysis on social media. The model is implemented through ontologies to explicitly and formally represent the knowledge related to OSNs. The detailed representation allows constructing automated analysis methods, and formalism provided through ontologies facilitates the correct interpretation and validation of the results obtained through automated techniques. A detailed explanation of the formal knowledge model is provided in [43].

A forensic investigation process model for semi-automated collection is designed. This model allows the automation of several processes and activities for forensic collection. The significant contribution of the model is to provide a way to identify the boundaries of data collection from a distributed and infinite medium, such as social networks. This model expresses the forensic limits through suitable parameters to facilitate the automated collection iteratively. A detailed explanation of this model is given in [44].

### 4.1. Data Collection

A Twitter scraper is used to automate data from online profiles. This scraper collects the data from online accounts using crawling behavior and Twitter API. The underlying scraping code that is coded in Python is obtained from an online source at (“GitHub-bpb27/twitter_scraping: Grab all a user’s Tweets (and get past 3200 limits)”, n.d.). However, the code is modified for making the metadata suit the requirements imposed by this work. The scraper code downloaded the raw data in JSON files, as indicated in Figure 4.

There are many forms of data retrieval from social media networks, including operator-based data retrieval.

### 4.2. Recovering Operator Data from Social Networking Websites

This form of data retrieval offers the possibility of recovering information or evidence about the operators who visit a profile on social network websites. These websites include integrated applications that display the operator names of people who visit any profile. Other sites keep log records that contain session information. Social snapshots are a novel way to collect digital evidence from social networking sites. PHP was used to retrieve the data. Some PHP functions and their applications are listed in Table 1.

### 4.3. Data Retrieved by Application Programming Interface API

An API enables various software programs to interconnect with one another.

Facebook’s API: Third-party Facebook applications can use the Graph API by writing JSON scripts and storing them in a separate file. The Facebook API includes the following features:The ability to find a list of acquaintances.The ability to display photos of acquaintances who are fans of a kind of story or article.An API explorer allows one to see associations between graph data and collect information for a beat or a specific story. JSON codes for API data reclamation in Facebook to retrieve plenty of acquaintances from an operator’s acquaintances list, as shown in Figure 5.

Twitter’s API: Twitter API, which is well documented and a suitable abode of helpful functionality, can be used to search for a specific term in Twitter and parse the results. The PHP codes for data recovery in Twitter are listed in Figure 6.

### 4.4. Ontologies and RDF Data Stores

A hybrid data model is implemented through ontologies and RDF datastores to facilitate detailed data and metadata storage and retrieval. This model consists of two-level ontologies. The lower-level ontologies represent local schemas or distinct social network platforms such as Twitter or Facebook. In contrast, upper-level ontology or high-level ontology implement forensic concepts. The upper ontology is also used to integrate multiple OSN data sources. The details of this model can be found in our previous work [44].

### 4.5. Mapping and Instantiation

The next step after data extraction is to populate the ontology using the result of the extraction process. Events, subjects, and objects are created from the extracted data to populate the ontology and corresponding RDF files. The associations among events, sub-events, subjects, and objects are deduced from the type of objects created. A detailed explanation of this process and ontologies is given in a previous publication [45]. The following constructs represent the example of Turtle serialized data representation in the data model and RDF stores. An example of serialized objects saved in RDF data stores is presented in Table 2.

### 4.6. Knowledge Enhancement

After instantiating and populating the data in ontology, the next step deduces knowledge from the extracted data. This stage is mainly valuable to improve the analysis steps’ results as it helps filter the most relevant knowledge about the entities and may discover some new knowledge. For instance, the investigators will be more concerned to gather evidence of defamatory material posted by the suspect. Hence, they can sort the events the suspect initiated or participated in by only using subject and object correlations, explained by the formal knowledge model.

These events can be filtered to select those events where the victim is mentioned or where both the suspect and victim have participated. According to the case instant under consideration, the content is examined for illegitimate and participants (subjects). The subjects encouraged the content and involved in the redistribution of the content are identified.

This process will significantly eliminate irrelevant events and deduce new knowledge, such as identifying new subjects or suspects in this case. This step is also considered helpful to narrow the focus of the investigation on a few specific subjects and objects from the bulk of data. This deduced knowledge is used for timeline construction and analysis.

### 4.7. Knowledge Extraction

Investigators can now pose high-level abstract queries on extracted content. For instance, they want to find the timestamps of all the content posted by a user on various social media sites. The query mentioned above is first written for global ontology using unified terminologies. Then, this query is reformulated for Facebook and Twitter ontologies separately to compute the results from distinct sources. Eventually, the mediator will combine the results from separate sources and present them in an integrated manner by using the consistent vocabulary of global ontology. An example of such queries is given in Table 3.

### 4.8. Automated Analysis Operators

In this article, we propose automated analysis operators to facilitate the ultimate goal of automated analysis. The general analysis encompasses a brief overview of the collected data, and the investigators can further choose to analyze a particular aspect in more detail. These operators are based on these measures; first, the relatedness among two incidents due to the same subjects involved. Second, due to shared or exchanged objects; third, temporal closeness or recurrence; fourth, geographical closeness. A list of proposed operators and their purposes is provided in Table 4.

These operators are only possible after realizing the previously presented semantic data and knowledge models [44,46]. The suggested formal knowledge provides the theoretical foundation for deriving these operators. The implemented semantic data model allows persistent storage and high-level data query to implement these analysis operators. These methods are used to find the correlations and relatedness among subjects and objects and organize the data in separate views, giving the investigators insight into the actual data. In current work, these operators are used for processing the data to categorize and filter the data for each view and present them through suitable and easy to understand visualizations. More importantly, the results produced by the proposed methods are logically explainable through the knowledge model; hence, they are justifiable for legal acceptance.

### 4.9. Visualizations

Visualizations are used to present the processed and filtered data because, occasionally, the textual and tabular data do not immediately reveal the trends. In this work, suitable visualizations are used to instantly highlight the most critical data for examination. For instance, the word clouds may highlight the most frequent words in the selected communications; moreover, the word cloud increases the size of words with higher frequency in the depiction. Network graphs represent the analysis to show social and interactive graphs. Scatter charts are also used to present the temporal activity patterns. However, it is imperative to experiment with several more visual representations to find the most suitable depiction for each operator or aspect of data, such as geographical patterns or presenting behavioral elements.

## 5. Case Study

This work is tested using a hypothetical case study, like several other digital forensic domain studies. Notably, evaluation by case study allows creating a complete and complex picture of whole phenomena, as stated in [46,47,48]. This work follows the theoretical approach for constructing case studies as presented in [49,50]. Christopher Galbraith evaluated his work by comparing the data of 28 individuals over seven days on the known same source and known different sources [49]. Hyunji Chung et al. created a hypothetical crime scene for investigation on cloud forensics. This scenario involved the leakage of a sensitive document by an employee and performed the necessary steps involved in the process to leave the traces on the network and machines that were collected and analyzed to identify the evidence of crime [50]. A similar approach is followed for constructing the case study to evaluate the proposed work presented in this article. A hypothetical case of cyberbullying is devised. A group of individuals carried out the necessary activities on the Twitter platform. The data were collected and analyzed for crime investigation and evidence identification [50].

In this work, cyberbullying is chosen as an instance for a case study. “*Cyberbullying*” or cyber-harassment refers to an individual’s offending behavior towards another person through the internet, mainly using social media sites. The bullying behavior may include posting rumors, threats, sexual remarks, a victims’ personal or sensitive information, or harsh labels such as hate speech [51]. Legally, bullying, or harassing behavior is identified by the offender’s repeated behavior and an intent to hurt [52].

### Test Data

The testing team created 26 new accounts on the Twitter platform, with unique names, in alphabetical order. The “TweetDeck app” is used for creating and managing the required number of Twitter accounts. TweetDeck is a dashboard application to manage several Twitter accounts simultaneously. However, in this scenario, for emulating the six key users’ behavior, including suspect, victim, and two close friends of each user, we needed individual people to manage these accounts separately for five months. Therefore, a team of nine people was set to help in this case study on a volunteer basis. Five of these nine people are from the same age group, 30–35 years old, and have the same educational level because the scenario indicates bullying behavior among peers. However, these five people have distinct daily routines.

## 6. Experiments and Results

This work has proposed and implemented a few automated analysis methods. OSN content is already arranged in chronological order; hence, most analysis methods further analyze time series data and the semantics of interactions among subjects through digital objects. The specific and casual correlations among subjects or objects can be highlighted per the investigation requirement. Identifying these associations and correlations depends on the case parameters given by the investigators to define the incident. The investigator specifies the incident, which outlines the scope of online automated data extraction from the OSNs. The investigators specify the period surrounding the auspicious event with a start date and end date and the subject’s name (suspect, victim, or witness). Furthermore, the investigator states the object type for collection that seems more relevant to him, such as tweets, comments, behaviors (like, share). Then, the automated extraction module for the specified SN platform will gather the data according to the given parameters and store it in the Jason file.

A detailed description of the terminologies such as subjects, objects, and evidence are described in our prior work [45].

This section presents the experimental results of implementing the analysis operators. These operators are executed on the semantic data model; the semantic queries extract the RDF stores’ data. These analysis methods use semantic queries and necessary mathematical and logical computations to quickly sort and filter the most relevant information from the extracted data. Two examples of these queries are shown in Table 5.

These queries are executed on the RDF data stores used to store the Twitter network’s data.

### 6.1. Interaction Graph

The study presented by Puttaswamy et al. explains the interaction distribution among the social graphs [53]. This operator helps to sort the contacts among the social graph on a social network. The sorting criteria are the number of interactions shared among users. Figure 7a,b and represent a directed and weighted interaction graph.

The figure shows the direction of interaction from the subject and towards the subject. The width of the lines and the distance from the center show the interaction frequency.

The interaction graph *IG* is a subset of the social graph. It is based on the frequency of interaction between the person of interest x and his/her social network contacts. The interaction graph is defined by two parameters, *fn* and *t*. Here, *fn* refers to the minimum number of interactions, and *t* is the period in which these interactions must have occurred. Together, *fn* and *t* define an interaction rate threshold. Hence, all the nodes (i.e., friends, followers) in the interaction graph are higher than the minimum interaction rate threshold. A similar approach is also observed in another work [53].

IG ⊆ SG.

The interaction graph is the set of users *u* who must have interaction rate fn, with the subject S, higher than the minimum interaction rate.

IG = {u_1_,u_2_,…….u_n_ | fn (u_i_) > min (fn)}

Where i varies from 1 to n i = {1, 2,….n}.

#### Frequency of Interaction

This operator is formulated to find the frequency of interaction among two users. It is already explained that sorting and filtering communications among users help identify their relationships’ dynamics. These two correlations describe the relatedness among the subjects due to shared objects.

The connectivity between two subjects, s_1_ and s_2_, rises correspondingly with the increase in the number of objects they exchanged or used for communication. These objects may include liking behavior, a comment, or a re-shared post. The *connectivity* is determined by the cardinality of these shared objects, as given below.
$$connectivity\;(s_{1},\;s_{2})=\frac{\left|Os1\cap Os2\right|}{max\left(\left|Os1\left|,\right|Os2\right|\right)}$$

|S_e_| and |O_e_| represent the cardinality of subjects and objects.

Figure 7a,b show directed and weighted communications among the subject and her contacts on the Twitter platform. Figure 7a shows the subject’s top twenty contacts most interacted with on OSN. The broader lines in the charts show a higher number of interactions. Figure 7b indicates the filtered and sorted data based on other users’ interactions with the subject.

The objects considered as interactions are the tweets, retweets, replied tweets, quoted tweets, user mentions, and direct messages. This step significantly reduced the amount of data collected in OSN investigations. First, it reduced data collection by sorting and filtering based on communication. Secondly, it limits the collection to specific objects. Those tweets are collected that are replied by or replied to the subject Alice. It also includes the tweets that have mentioned the subject. This kind of information is retrieved using a query as given in Table 5.

### 6.2. Temporal Activity Graph

This operator is used to analyze a user’s activity pattern concerning time. Analysis of time series data can show several valuable statistics and expressive features of the data. Several studies have reported relationships in temporal patterns and online activities. Even regular weekdays and weekends show a change in online activity concerning time. Additionally, the timings of using mobile phones may represent a person’s sleep and wake patterns [54,55]. This information is valuable in recognizing time zones and estimated geographical locations of an individual uploading the content on OSNs. Typically, the phone using time patterns and temporal association of online activity is unique for each and tends to repeat [56,57]. Therefore, examining and comparing an anonymous person’s repetitions with a known individual may help identify the anonymous user. We explain the few assumptions related to the time of the activities outlined below:Repetition of online communication at a particular time exposes a relatedness among the objects or subjects, *pattern (S*_1_*, S*_2_*),* and outlines an individual’s routine in terms of online social activity.The relatedness among two activities and subjects can be represented by the relatedness of two different online activities. If two activities, *a*_1_ and *a*_2_, occurred in a period from *t*_1_ to *t*_2_, the *interval* is given by *(a*_1_, *a*_2_, *t*_1_, *t*_2_).

The timestamps are collected and associated with each person’s activity in a specific interval and are plotted on charts with dates and times, as shown in Figure 8.

### 6.3. Tweet Cloud

This operator is formulated to give a quick overview of the words and concepts dominant in someone’s tweets; as it is a time-consuming procedure to separate the tweets of an individual user and read them exclusively. Therefore, this operator is crucial as it reduces the time and effort needed to understand the subject’s content.

It is important to note that this operator is not only used for summarizing the tweet content, but it also calculates the dominance of concepts by calculating the number of times a word appears in collected tweets. It further allows filtering the concepts by using the keyword or setting a threshold based on the number of terms.

The text of all the suspects’ tweets in the given time frame is extracted. A word list is generated for those tweets by counting each letter’s number of times in the tweets. The frequency of appearance is assigned a weight to each word. The word list is presented by the word cloud to increase comprehension. The word with higher weights receives a new visualization. This visualization will show fewer words to obtain a more precise idea of the tweets. An example of a tweet cloud generated from the tweets of a cyber-bullying suspect is shown in Figure 9.

### 6.4. Hashtag Cloud

This operator is designed to give a quick overview of the hashtags used in someone tweets. It is just like a tweet cloud.

### 6.5. Similarity of Views

This operator is based on the sharing feature of Twitter and is used to identify the similarity of opinion among two users. The number or frequency of interactions cannot check the similarities and differences of thought, so the data analysis also verifies this. This operator is based on a rule-based correlation. This operator separates and sorts the content that is re-posted by the user. If user A re-posts content shared by user B several times, it indicates that user A likes or agrees with that content enough to share it with his other friends.

Let O_1_ and O_2_ be the set of all objects created by subjects S1 and S2. O_S1_ and O_S2_ are the sets of objects shared by S_1_ and S_2,_ respectively. Likewise, O_R1_ and O_R2_ are the sets of objects re-shared by S_1_ and S_2,_ respectively
O_R1_ ⊆ O_S1_ ⊆ O_1_
O_R2_ ⊆ O_S2_ ⊆ O_2_

Hence, the endorsement of the views of S_1_ and S_2_ is given by (1), and the approval of S_2_ by S1 is given by (2).
$$\mathrm{Sim}\;(S_{1},S_{2})=\frac{\left|O_{s1}\cap O_{R2}\right|}{\left|O_{S1}\right|}$$
$$\mathrm{Sim}\;(S_{2},S_{1})=\frac{\left|O_{s2}\cap O_{R1}\right|}{\left|O_{S2}\right|}$$

The (1) and (2) values might not be the same, indicating a partial agreement among the subjects. However, a significant difference between the two values may mean a social distance or influence of opinion.

There are two types of directional interactions. The first is “By _Subject”. These are the communications sent or posted by the subject—Alice, in this case. The other interactions are the communications sent to, mentioned, or replied to the subject by her friends/followers. We used the number of retweets to measure the agreement between the subject and her contacts. Retweets refer to the tweets posted by others and retweeted by the subject. Retweeting conduct indicated a positive or supportive behavior towards the content being re-posted. Figure 10a represents the rate of tweets by subject, retweeted by her contacts. In comparison, Figure 10b indicates the retweets by subject and the users whose tweets are retweeted by subject.

### 6.6. Geo-Location Activity Graph

This operator helps sort the locations that are tagged in online content. We collected the coordinates and places found in the metadata that accompanied the subject’s tweets and plotted the positions using Google Maps visualizations, as shown in Figure 11.

An example of the query for retrieving geographical data from RDFstores is given in Table 5 query B. (B).

### 6.7. Trace Operator

At the end of every investigation, investigators must select the evidence for presentation in court. They will have all the associated data with the offending comments. In addition to the metadata (i.e., time, device, platform, profile data), they have data related to other subjects (i.e., Dave, Eve) who have witnessed the illicit activity and participated in that event. Links from evidence to the objects are shown in Figure 12.

All these data will help to corroborate the evidence in court. Furthermore, the investigators can explain the process from the first step to the end. Evidence must be related to an entity en ϵ {E × S × O}. Here, *E* represents the event, S is for subjects, and O refers to objects defined by the incident.
Trace (en ϵ {E × S × O}) = {Ev ϵ Tr | even σ_t_ en}

Trace is an operator that links the evidence to the entity. The Figure 12 shows how the final results, as displayed in time–activity pattern charts, are related to the data model, and then they can directly be traced to raw data.

## 7. Limitations

This work is tested and evaluated by using a hypothetical case study. We created a dataset on the Twitter platform by emulating a case of cyber-bullying. This dataset involves several users who participated in creating a said dataset for a period of 3 months. These data helped us test and evaluate the semantic data model and test the proposed automated operators on the dataset like actual data. However, this dataset is smaller than the real-world data of online social networks, although this work does not evaluate the proposed solution’s efficiency and scalability.

Several factors compelled us to create our datasets, such as a lack of available datasets in the digital forensics domain and online social networks’ privacy issues. According to a Twitter development agreement, which must be agreed upon before accessing data, the developers cannot share complete datasets. Their guideline stated that the developers could not share any content or downloadable datasets with others; they can only share tweets or user IDs or share data by anonymizing it [58].

Therefore, several other researchers in the domain are using their own datasets for experimentation. The work presented by Mohammed Ali Al-gardai et al. used a dataset with metadata and other features of activity and network data on Twitter to identify cybercrime and cyber-bullying [59]. In a study based in Malaysia, Balakrishnan et al. demonstrated that cyberbullying increases directly with increased online activity [60]. They also created their own dataset. According to the findings, Internet frequency was found to significantly predict cyber-victimization and cyberbullying, implying that as time spent on the internet increases, so do the chances of being bullied and bullying someone. Therefore, when analyzed and found to be accurate concerning the problem under investigation, online information can be used as a criminal case study.

## 8. Conclusions

This work explains a model that allows the extraction of essential data from online social networks and prepares them to be presented in legally acceptable formats. This article presents the proof-of-concept implementation and results of automated analysis operators that can be integrated into software tools to assist in analysis and processing.

This work explained the feasibility of using a multi-layer semantic approach for automated online social network forensics techniques. Currently, computerized techniques are needed to manage the massive volumes of digital data and their complexity on online social networks. Otherwise, the investigation process would become inefficient. The inability to deal with data complexities might lead to errors in the process that would severely affect the subjectivity. Currently, most OSNs are implementing the techniques to identify hate speech and abusive language. However, this will not affect the significance of the proposed work because it is not limited to identifying hate speech or cyberbullying, although it was used as a case study. Word-based tweet clouds are used in only one analysis operator. As rest of the operators show other aspects of data, such as temporal activity graphs extracted from the timestamps attached against each activity in metadata. Social interaction frequency and a user’s bias toward a specific idea or person are extracted from the online engagement behavior of the user. Likewise, the user pattern of visiting physical places is also extracted from metadata or OSN tags.

The proposed work presents a forensic investigation model and practices a few analysis operators to perform advanced analysis on the OSN content. The trace operator describes a practical and automated method to establish and prove the provenance of entities that lead to evidence. Therefore, this work has suitably proposed a computerized approach for OSN forensic analysis and provided a proof-of-concept implementation. Additionally, it gives the fundamental requirements for implementing automated solutions in other domains such as cloud or IoT forensics. This work uses the Twitter platform to demonstrate the feasibly and practicability of the proposed approach. In this regard, any other social media platform can be used to collect test data, although Twitter API is easy to use and provides complete and free access for academic use. Platform-specific data extractors are used for collecting the data from each OSN. Each platform provides its APIs to extract data; hence, it is impossible to have a single extractor to extract data from all the platforms. Likewise, a platform-relevant parser is used to translate the data to OSN ontology. Once the data are integrated into the semantic data model, analysis operators can use them collectively. The future research will expand this study by involving other social media network databases such as Facebook.

## Figures and Tables

**Figure 1 sensors-22-01115-f001:**
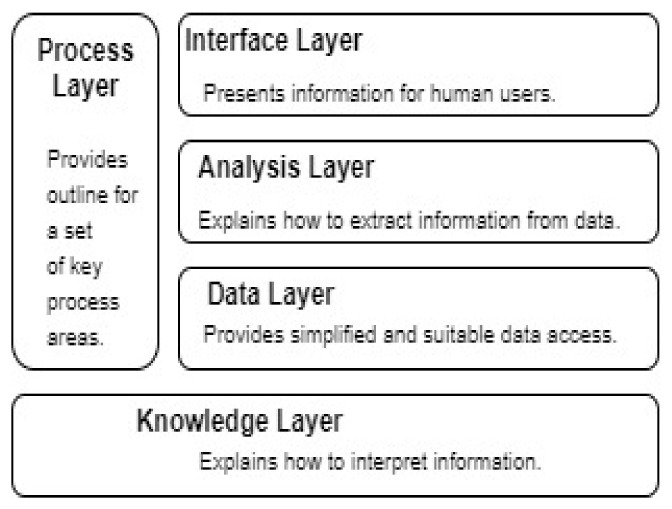
The multi-layered conceptual model.

**Figure 2 sensors-22-01115-f002:**
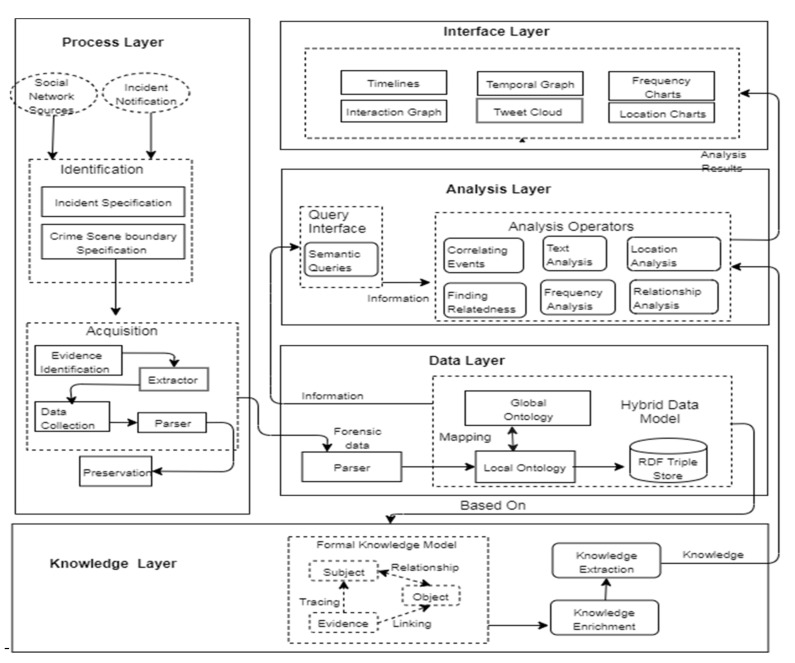
The architecture of the multi-layered model.

**Figure 3 sensors-22-01115-f003:**
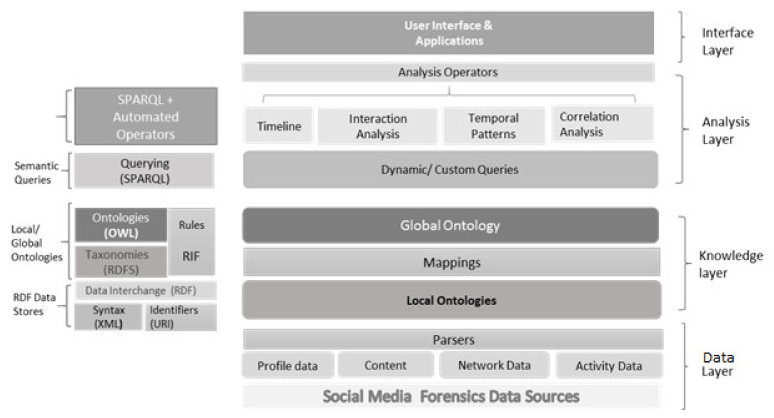
Implementation details of the multi-layered model.

**Figure 4 sensors-22-01115-f004:**
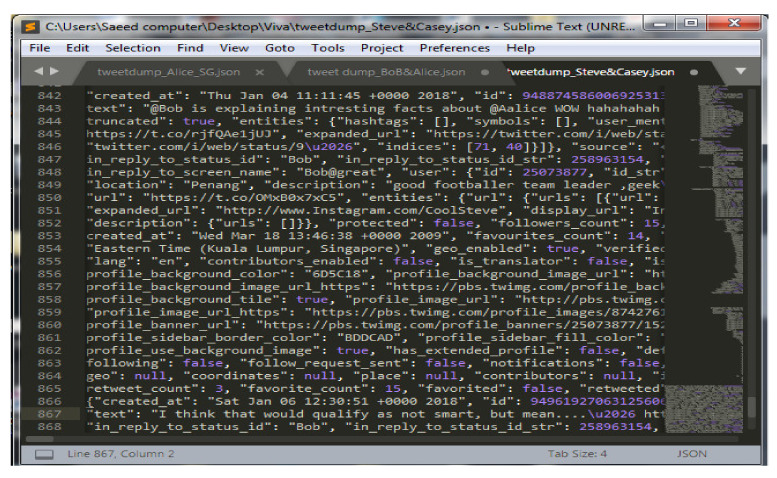
Raw data downloaded by Twitter scraper.

**Figure 5 sensors-22-01115-f005:**
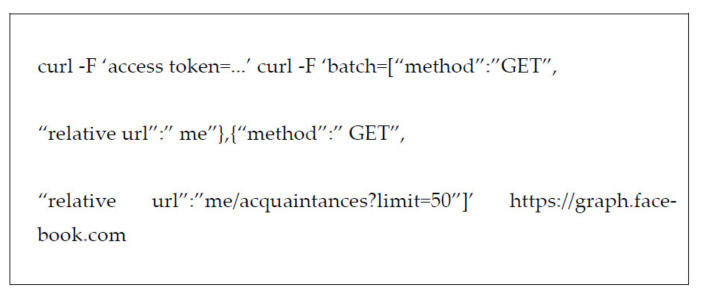
PHP codes for Facebook API data retrieval.

**Figure 6 sensors-22-01115-f006:**
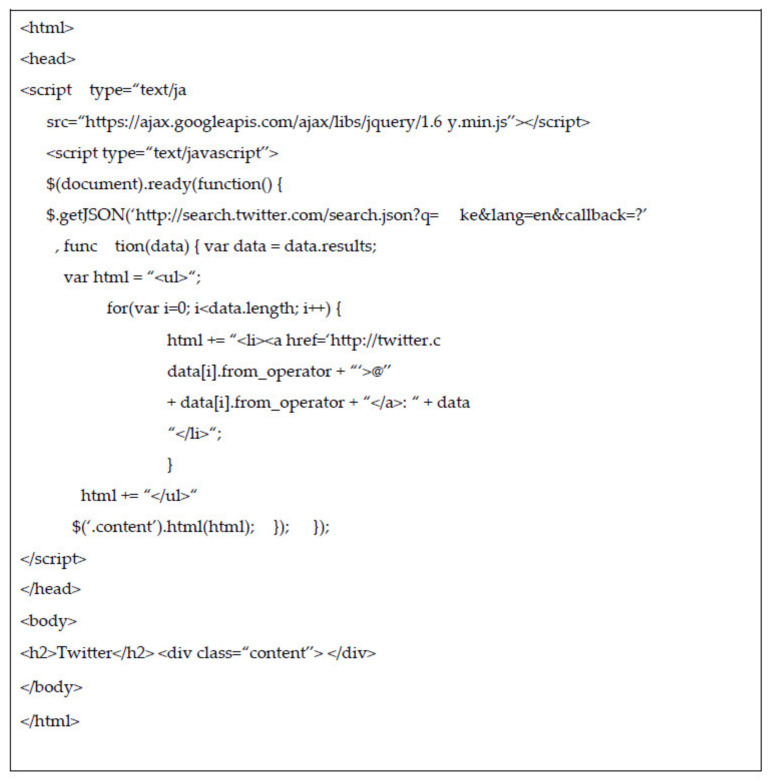
PHP codes for Twitter API data retrieval.

**Figure 7 sensors-22-01115-f007:**
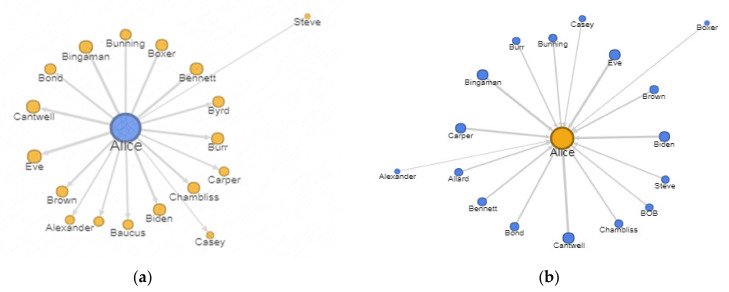
Interaction graph (**a**) from the subject to her contacts; (**b**) from contacts to subject.

**Figure 8 sensors-22-01115-f008:**
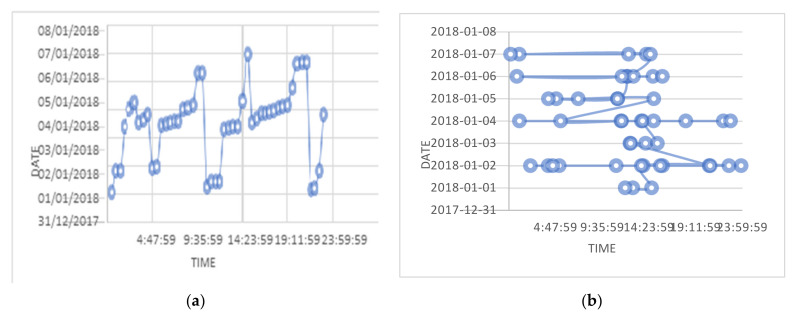
(**a**) Temporal activity graph for the subject. (**b**) Temporal activity graph for one of the users.

**Figure 9 sensors-22-01115-f009:**
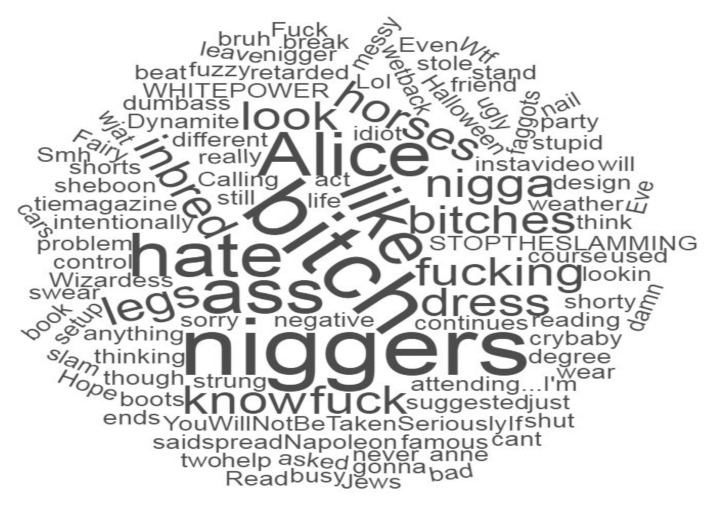
Tweet cloud generated from the tweets of a cyber-bullying suspect.

**Figure 10 sensors-22-01115-f010:**
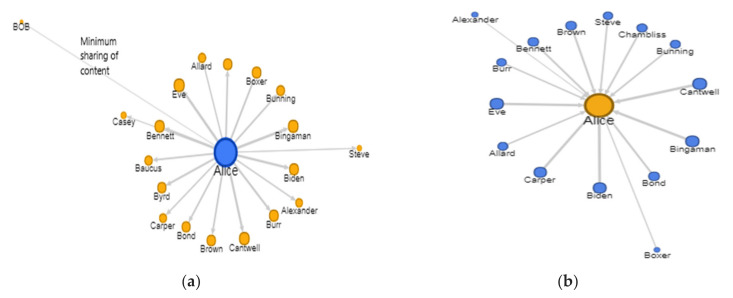
(**a**) The objects of the subject re-shared by her contacts. (**b**) The objects of other users re-shared by the subject.

**Figure 11 sensors-22-01115-f011:**
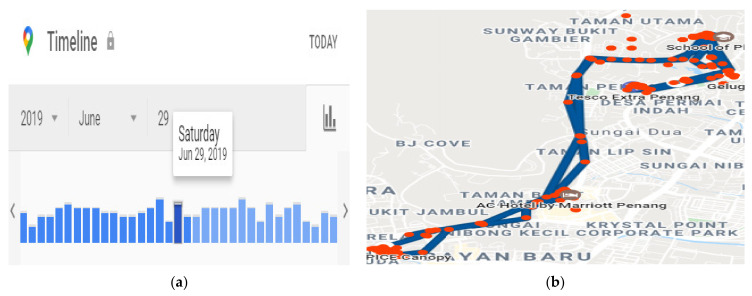
(**a**) Timeline of places visited by subjects. (**b**) Raw geographic data.

**Figure 12 sensors-22-01115-f012:**
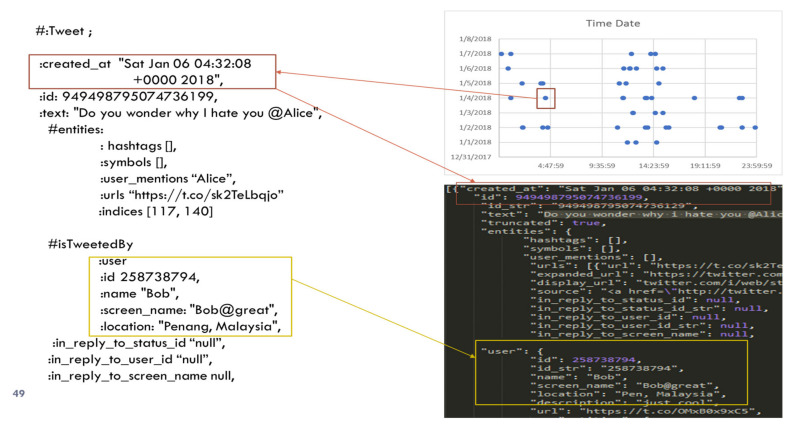
Links from evidence to the objects.

**Table 1 sensors-22-01115-t001:** PHP functions and their applications.

PHP Functions	Applications
getmxrr ($hostname, $mxhosts)	Obtain the names of mail exchanger hosts for a specific host.
gethostbyaddr ($ip)	Collect the hostname associated with an IP address.
gethostbynam e($name)	Gather the IP address associated with a hostname.
checkdnsrr ($host, $type)	Checks the DNS for records of type $type for host $host and returns Boolean true if any are found.
dns_get_record ($host, $type)	Retrieves the DNS record for host *$host*. The $type parameter, on the other hand, is an option.
getservbyname ($service, $protocol)	Obtains the port number for the service $service via the protocol $protocol.

**Table 2 sensors-22-01115-t002:** Turtle serialized data representation of *USER* and *TWEET* construct in the data model and RDF stores.

User Concept on Twitter.	An Individual of Class Tweet.
**#User** **:** **User an owl: Class ;** **:Bob a owl:NamedIndividual,:User;** **:name “Bob”;** **:user_id “258738794”** **:profile data** **:created_at “Sat Jan 06 04:32:08 +0000 2018”;** **:favourites_count “100” ;** **:followers_count “50” ;** **:friends_count “20” ;** **:statutses_count: “20”;** **:geo_enabled “true”;** **:contributors_enabled “false”;** **:has_extended_profile “false”;** **:id_str “25073877”;** **:location “XYZ”;** **......** **…**	#Tweet : Tweet an owl Class : tweet013 a owl: NamedIndividual: Tweet ; “created_at”: “Sun Jan 07 04:32:08 +0000 2018”, :id: 949498795074736199, :text: “Do you wonder why I hate you @Alice”, #entities: : hashtags [], :symbols [], :user_mentions “Alice”, :urls “ https://t.co/sk2TeLbqjo (accessed on 23 August 2021)” :indices [117, 140] # isTweetedBy user: :id 258738794, :name “Bob”, :screen_name: “Bob@great”, :location: “Penang, Malaysia”, :in_reply_to_status_id “null”, :in_reply_to_user_id “null”, :in_reply_to_screen_name null,

**Table 3 sensors-22-01115-t003:** SPARQL queries to extract data from RDF stores.

SELECT ? time_stamps WHERE{ ?user rdf:type snfo:subject. ?user efiosn:name “Alice”. ?objects efiosn:isCreatedBy ?User. ?objects efison:creation_time ?time_stamp. }	SELECT ? time_stamps WHERE { ?user rdf:type twitter:User ?user twitter:name “Alice”. ?tweets twitter:isTweetedBy ?User. ?tweets twitter:createdat ?time_stamp. }
Query (A). Timestamps for all the objects created by a user “Alice.”	Query (B). Timestamps for all the Tweets Created by a user “Alice.”

**Table 4 sensors-22-01115-t004:** List of Analysis Operators.

	Operators Name	Description
1	Interaction Graph	This operator uses subject and object correlations. It helps to sort the contacts among the user’s social graph, with the highest frequency of communications.
2.	Interaction Frequency Analysis	This function is based on subject and object correlations. It is used to perform a frequency analysis of communications among two users to sort and filter communication among users. It helps identify the dynamics of their relationships.
3.	Temporal Activity Graph	This function uses temporal correlations, as explained in the section. It is used to analyze a user’s activity pattern in a specific period.
4.	Geo-location Activity Graph	This operator uses object correlations and helps sort the locations that are tagged in online content.
5.	Hashtag Cloud	This function is based on object correlations and is designed to give a quick overview of the hashtags used in tweets.
6.	Tweet Cloud	This method is also based on object correlations and is designed to give a quick overview of the topics or themes existing in someone’s tweets.
7.	Similarity of Views	This operator is based on rule-based correlations and identifies the nearness of opinion among two users.
8.	Trace Operator	Trace is an operator that links the evidence to the entity.

**Table 5 sensors-22-01115-t005:** SPARQL queries are used to retrieve data from RDF data stores.

**(A)** **A query executed on the RDF Model using Twitter ontology to select the tweets tweeted by Subject (A) and retweeted by Subject (B).**	**(B) ** **A query is used to extract the places tagged by the user in her retweets, reply tweets, and quoted tweets.**
**Select ?retweets ?id** **where {** **?x rdf:type twitter:User.** **?x twitter: name “Alice”.** **?x twitter:user_id ?uid.** **?tweetby rdfs:subPropertyOf** **twitter:isTweetedBy.** **?tweets ?tweetby ?x.** **?y rdf:type twitter:User.** **?y twitter: name “Eve”.** **?y twitter:user_id ?uid.** **?tweetby rdfs:subPropertyOf** **twitter:isRetweetedB ?retweets ?isRetweetBy ?y.** **?retweets twitter:id ?text.** **}**	Select ?loc ?placename ?lat ?longt ?time where { ?x rdf:type twitter:User. ?x twitter: name “Alice”. ?x twitter:user_id ?uid. ?tweetby rdfs:subPropertyOf twitter:isTweetedBy. tweets ?tweetby ?x. ?tweets twitter:isTaggedWith ?loc. ?tweets twitter:created_at ?time ?loc twitter:place_name ?placename. ?loc twitter:latitude ?lat. ?loc twitter:latitude ?longt. }
Result: A List of retweets and their ids tweeted by Subject(A) and retweeted by Subject (B).	Result: A list of Place names, longitude & latitude values.

## Data Availability

This article contains no data or material other than the articles used for the review and referenced.

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
