# Peer review of "A Multi-Layer Semantic Approach for Digital Forensics Automation for Online Social Networks"

_sensors, 2022, doi:10.3390/s22031115_

Round 1

Reviewer 1 Report

Over time, theoretically and practically, social networks have demonstrated a serious field of data used to determine user behavior. In terms of determining the degree of crime, social networks become limited if private conversations are taken into account (due to p2p encryption).  After 2013, with Snowden's revelations, social networks and other actors refused to provide data without warrants. The present study implies serious limitations when it comes to extracting concrete data and, also, is unable to obtain private data (messaging conversations). Overall, the model presented can become an essential tool if it is used following a court decision that would cause the service provider to make the data available to investigators.

Reviewer 2 Report

The research objective is interesting and seems to answer an important and urgent problem. However, the study shows gaps both in the methodological approach and in the readability of the text which is not clear in all its parts.

With regard to the literature review, many of the cited studies are somewhat outdated, given the ever-evolving research field.

In many passages, the method of recalling the studies (with the sole indication of the reference number from the final list) does not make the understanding of the periods fluid and complete. It is necessary to report the key elements referred to in order to prevent the reader from continuously interrupting the reading.

The definition of "Social media forensics" in section 2 results unclear and should be further investigated. The referenced subset is not explicit. Bibliographic references in this part of section 2 are missing.

Issues related to privacy are, by explicit admission of the authors, fundamental in the design of a digital forensic automation system. However, this question is never explicitly addressed. It is not clear how privacy issues are handled in the model. Who are they responsible for (OSN providers?)

Throughout the document, there are too many references to other studies, even by the authors themselves, which make it difficult to understand. For example, in lines 386-387, authors said:

“This work follows the theoretical approach like [50-51] and using the dataset construction methodology like [52-53]”.

 It is necessary to explain how the hypothetical case study was set up by summarizing the key elements of the approach used, otherwise the period is incomprehensible. The same problem is encountered with the following sentence (lines 412-414):

“The details of specifying the incidents for automated analysis and the formal theory to explain the association among subjects, objects, and evidence are described in our prior work [45]”.

In section 3, the approach described refers to the logical and IT development of the model. However, there is no explanation of the methodological approach to the research design, of which the model is the product. How does one come to establish that a multi-layered model is the right solution to the problem presented? It is suggested to review the methodology in the light of an approach such as the Design Science or similar. In any case, it is necessary to deepen and explain what is said at the beginning of section 3: “This study examines the potential areas of a forensics investigation where automation can be applied without contradicting online social networks' legal and privacy requirements (linee 208-209)”.

Regarding the applicability of the model, the motivation for choosing Twitter for implementation is lacking. Does it depend on API availability or other factors? Clarify.

Improve the correspondence between the labels encoded in Figure 3 and those used in the text to explain the implementation of the model. Use the same terminology especially in the paragraph headings to help the reader understand the various steps of the model.

Regarding the implications, considering the technological and policy differences between the various social networks, how can this automation model be applied to other social networks? The choice of verifying the applicability of the model through Twitter seems to limit the applicability of the model to other social networks. It should be explained how the model can be adapted to other social networks. Does it involve a change in the source code? Paragraph 4.3 seems to want to answer this question but it does not exhaust it. Also because from par. 4.1 only the specificities of Twitter are introduced. In 4.3 only the Facebook API is mentioned in addition, before returning to Twitter.

An additional suggestion: Nothing was said about the fact that by now all OSNs, including Twitter, already implement systems of moderation of content that incites hate, which incorporates words / expressions to be silenced and systems of reporting harassing accounts. Does this aspect impact the usefulness and functionality of the model? It is suggested to face and clarify this point both in terms of implications and to suggest future developments of the model in the light of the evolution of social networks in this direction.

Reviewer 3 Report

The research focuses on the critical topic of automating digital forensics, as well as technical and legal concerns. The work is well-presented and will pique the readers' interest. There are some suggestions to improve the work further:

  1. The introduction is adequate, but could be improved. The gaps in the state of research at the moment are not entirely clear. The authors should make an attempt to address it near the conclusion of the introduction.
  2. The authors should refer a bit more. Here are few examples:

(a) Section 1.2.1 – reference should be given while defining data provenance (lines 76-77).

(b) Section 2 - Provide reference to attacks and threats on social media sites (lines 102-103).

(c) Section 2 - Provide reference to DoS, DDoS occur outside the system (lines 105-106).

(d) Section 2 – Provide reference to “Researchers are now studying the point of vulnerability of these social networking sites, a process known as Social Network Forensics.” (lines 107-109)

  1. There are sentences that need clarification on language, like

(a) Section 2 – “Existing approaches exist” (line 135)

(b) Section 2 – “However, they insist” (line 203). Clarify who insist.

  1. At the conclusion of section 2 – related works, a case for automation is made. However, in this section, please make a case for why current research on automation is falling short and how this study will address these gaps.
  2. The study's implications should be discussed in greater detail in the conclusion, including both technical and legal implications.
